# Effects of Various Drying Times on the Properties of 3D Printed Orodispersible Films

**DOI:** 10.3390/pharmaceutics14020250

**Published:** 2022-01-21

**Authors:** Natália Janigová, Jan Elbl, Sylvie Pavloková, Jan Gajdziok

**Affiliations:** Department of Pharmaceutical Technology, Faculty of Pharmacy, Masaryk University, Palackeho Trida 1946/1, 612 00 Brno, Czech Republic; 507082@muni.cz (N.J.); elblj@pharm.muni.cz (J.E.); pavlokovas@pharm.muni.cz (S.P.)

**Keywords:** orodispersible films, 3D print, semisolid extrusion, drying time, moisture content

## Abstract

Orodispersible films are an innovative dosage form. Their main advantages are the application comfort and the possibility of personalization. This work aimed to evaluate the influence of different drying times on the properties of orodispersible films of various thicknesses, prepared in two different semisolid extrusion 3D printing setups. In the first experiment, drying times were dependent on the overall print time of each batch. In the second setup, the drying time was set equal according to the longest one. The evaluated parameters were films’ weight uniformity, thickness, moisture content, surface pH, disintegration time, hardness, and tensile strength. Upon statistical comparison, significant differences in the moisture content were found, subsequently affecting the disintegration time. Moreover, statistically significant differences in films’ mechanical properties (hardness, tensile strength) were also described, proving that moisture content simultaneously affects film plasticity and related properties. In conclusion, a mutual comparison of the manufactured orodispersible films showed that the drying time affects their physical and mechanical properties. The in-process drying setup was proved to be sufficient while allowing quicker manufacturing.

## 1. Introduction

Orodispersible films (ODFs) are single-layer or multi-layer thin polymer sheets intended for rapid dissolution or disintegration in the oral cavity. They are usually applied directly to the tongue (Ph. Eur.) [1]. The FDA defines ODFs as a thin dosage form that disintegrates rapidly upon contact with a liquid [2]. A suitable ODF should be thin and flexible but resistant enough to be easily used and simply packed. The film should not be sticky and hold the form without rolling. Another important aspect is the pleasant taste and low irritability. The disintegration time should be as short as possible [3]. Most of these properties are influenced by the content of water or other plasticizers in the film structure.

ODFs are still not a common dosage form. However, oral films can be found in patent literature from the 1960s. Pfizer was the first company to start manufacturing breath freshener Listerine^®^ PocketPaks in 2001 [4]. In 2003 the first oral film with API (Chloraseptic^®^ with benzocaine for sore throat treatment) was rolled out, followed by the first oral strip containing a prescription drug (Ondasetron Rapidfilm^®^, antiemetic), registered in 2010 [4,5].

ODFs, as an innovative dosage form, take advantage of simple application and rapid disintegration in the mouth, suitable for patients with difficulty in swallowing tablets (children, geriatric patients, patients with Parkinson’s disease, dysphagia, or vomiting). ODFs can impact drugs’ therapeutic effects [5]. They reduce side effects like gastrointestinal irritation. When the film is applied into the oral cavity, part of the active substance is absorbed directly into the bloodstream through the oral mucosa. Hence, acid degradation in the stomach may be limited, which is advantageous for drugs sensitive to low pH, and the hepatic first-pass effect is reduced, increasing bioavailability [5].

On the other hand, oral films are burdened with certain limitations, such as the ability to incorporate only a limited amount of drugs; hence they are mainly suitable for highly effective and low dose drugs. Their production requires the use of solvents and subsequent drying, which may adversely affect the stability of the drug [6]. The main hurdle for patients’ acceptance of ODFs is taste (most APIs are bitter or generally unpleasant). Usually, it is necessary to add sweeteners and flavorings to the composition to mask the taste of the active ingredients, which reduces the usable amount of API and may even have a deteriorative effect on the film-forming properties of the main polymer [4,7].

Oral films are produced in several ways. The most common is solvent casting and hot-melt extrusion. Other options are the rolling method or spraying the layers onto the first layer formed by solvent casting. The main disadvantage of these methods is the need for subsequential steps of drying and cutting to the desired shape and size [4,8].

In the last decade, the production of dosage forms using 3D printing has come to the fore. This technology enables on-demand fabrication of personalized medicines and customized dosage forms in a layer-by-layer fashion into the geometry represented by a digital model. By using 3D printing it is possible to create an individual medication with different release profiles, active substance content, or the shape and size of the dosage form quickly [8,9]. Presumably, 3D technology will improve the resulting complexity of the product and allow personalization and on-demand manufacturing since it brings in possibilities to create small batches at low cost, even in compounding pharmacies [10,11].

Several 3D printing technologies have already been applied in the preparation of ODFs. In 2019, personalized ODFs with warfarin were prepared by semisolid extrusion (SSE). The results confirmed excellent dose to dimension linearity and content uniformity, important for substances with a narrow therapeutic index. Since this is a one-step process, it also can be applied in on-demand hospital compounding [12]. Another example of 3D printed ODFs are films containing aripiprazole prepared by fused deposition modeling (FDM), containing polyvinyl alcohol (PVA) as the main film-forming component. The results show that this method is certainly suitable for preparing personalized dosage forms. FDM printing also incorporates specific features, such as haptic identifiers or Braille encoding for visually impaired patients [13].

Moreover, since no solvent is used, recrystallization issues, generally connected to SSE, are not a concern for FDM. It is even possible to enhance the solubility of the incorporated drug by solid dispersion forming through the process of hot-melt extrusion. However, FDM lacks the simplicity of SSE due to the extra work step of filament preparation and prepared ODFs suffer from prolonged disintegration time, inherent to FDM technology [14]. In another study, orodispersible films with maltodextrins were prepared by direct extrusion of melted excipients, omitting the preparation of filaments typical for FDM. Moreover, printing the ODFs directly onto the primary packaging reduced another work step through this approach [15,16].

The combination of 3D printing and orodispersible forms could be a step towards automation and higher efficiency in preparing drugs tailored to a specific patient in a hospital or pharmacy. An example is the formulation of ODFs with cannabinol, which may be dosed exactly according to the patient’s weight [4,17].

In all the technologies for ODFs preparation (solvent casting, SSE), where excipients and active ingredients are dissolved or suspended in a solution, drying is the critical step, as it may affect the physical state of the incorporated drug and the overall properties of ODFs [4].

When improper or uncontrolled drying is used, unstable or therapeutically unsuitable polymorphs may be present in final ODFs since drug recrystallization may be affected by the length of drying, temperature, ambient humidity, and solvent used. This may reduce treatment effectiveness or even result in adverse effects due to the presence of toxic polymorphs [18].

Regarding ODFs’ properties, the relation between residual moisture content (RMC) and one of the main parameters of ODF quality, disintegration time, was established by Preis et al. who concluded that films of higher RMC tend to disintegrate quickly [19]. Appropriate RMC is also crucial for plasticizing effects. The type and amount of plasticizers significantly impact ODF flexibility closely related to the ease of handling and packaging [4,20].

High RMC in ODFs can also lead to physical instability of the API (drug recrystallization) and to increased water activity, which facilitates microbial growth [21]. Borges et al. proposed a range of 3–6% RMC for ODFs based on evaluating commercial ODFs of various compositions [21]. In another study, Foo et al. manufactured orodispersible films with an RMC of 3.4–6.2%. Their mechanical and application properties were satisfactory [22].

This work aimed to compare two setups of drying used to prepare ODFs by SSE 3D printing. The selected formulation was used to prepare ODFs of different thicknesses by the SSE printing method modified according to the previous experiment [10], which was repeated to prove the reproducibility of the process. Subsequently, the drying process was modified to allow more thorough drying by equalizing drying time for all batches differing in film thicknesses. Statistical evaluation of film properties was undertaken to assess whether equalizing the drying time improves the properties of ODFs enough to justify prolonging the preparation by additional drying and to evaluate the effects of drying setup on ODFs in general.

## 2. Materials and Methods

### 2.1. Materials

Maltodextrin (Glucidex 6–G6), the film-forming polymer with a DE value of 5.5, was provided by Roquette (Lestrem, France). The plasticizer sorbitol (Sor) was purchased from Dr. Kulich Pharma (Hradec Králové, Czech Republic). The thickener hydroxyethyl cellulose (Cellosize^®^ QP300 (HEC)) was kindly donated by DOW Chemicals (Midland, MI, USA). Quality purified water, according to Ph.Eur., was used.

### 2.2. Preparation of Dispersion for 3D Printing

The dispersion for 3D printing was prepared by dissolving Glucidex 6 (8% *w*/*w*) and sorbitol (5% *w*/*w*) in purified water. This solution was heated to 80 °C. Subsequently, HEC (1% *w*/*w*) was added slowly under continual magnetic stirring. The dispersion was then cooled down to ambient temperature during stirring at 300 rpm (2 h). The amount of evaporated water was refilled [10]. Water content in the printing dispersion was 86% (*w*/*w*).

### 2.3. Preparation of Models and Printer Setup

Models for 3D printing were prepared in software Blender 2.78c (Blender Foundation, Amsterdam, The Netherlands). The basic models were composed of 30 rectangles having a 20 × 30 mm footprint. The model heights were 45, 85, 125, 165, and 205 µm in respective batches. Exported stereolithographic files (.stl) were sliced in Slic3r PE 1.33.8 (Prusa Research Ltd., Prague, Czech Republic).

A modified 3D FDM printer was used in both experiments, with the extruder replaced by a linear syringe dispensing system. This modification simplifies the dosing of dispersion as the original extruder motor controlled the syringe plunger. The syringe diameter was equivalent to the filament diameter parameter used by Slic3er software (Prusa Research Ltd., Prague, Czech Republic). The syringe volume was 50 mL and its internal diameter was 29.28 mm. The syringe was connected through a 50 cm tube (2.54 mm dia.) with a 0.84 mm diameter needle tip [10].

### 2.4. Printing Parameters

This experiment follows up on previously presented work using identical printing parameters shown in Table 1 [10]. The material was dosed in a 10-fold excess and the initial distance between the heated bed and syringe tip was set to 0.2 mm.

Experiments I. and II. varied in the drying procedure. In experiment I., the films were kept on the heated bed (70 °C) for an additional 10 min after printing. Therefore, the overall drying time (including in-process drying during printing) varied in every batch. In the second experiment (II.), the drying time was equalized (114 min) for all batches. Drying times can be found in Table 2.

### 2.5. Evaluation of Physico-Chemical Properties of ODFs

#### 2.5.1. Weight and Thickness of ODFs

The weight of films (*n* = 30) was measured by analytical scales KERN 440–445 (Gottl. KERN & Sohn GmbH, Balingen, Germany) [10]. The results are presented as mean values ± SDs.

Film thickness was measured by the coating thickness gauge Elcometer 456 (Elcometer Limited, Manchester, UK), based on the electromagnetic induction principle. The thickness of each film (*n* = 30) was measured at 5 points over the area of the film (4 corners and the center) [10]. Results are presented as mean values ± SDs.

#### 2.5.2. Moisture Content

Evaluation of moisture content (MC) was undertaken using a halogen moisture analyzer (Excellence Plus HX 204, Mettler Toledo, Greifensee, Switzerland), working on the thermogravimetric principle. Each evaluated sample was heated to a constant temperature of 105 °C. The measurement was finished after a stable weight was reached (weight change less than 1 mg over 50 s) [10]. The measurement was repeated five times; results are presented as mean values ± SDs. The RMC measurement was the last test. It took place about 2–3 h after the end of the printing itself. The films were stored in a sealed plastic container. They were exposed to air humidity for the necessary time in other tests.

#### 2.5.3. Surface pH

A contact pH meter (Flatrode, Hamilton, Bonaduz, CH) was used to evaluate the surface pH of films. A drop (0.1 mL) of purified water was applied to each evaluated film surface, and then the electrode was laid on the wetted film [10]. Results were read after 30 s. The measurement was repeated five times with results presented as mean values ± SDs.

#### 2.5.4. Disintegration Time

The modified disintegration tester equipped with the film holder clamps was used to evaluate disintegration time (DT) [10]. Each ODF was magnetically pinned by a 3 g weight, chosen to approximate the minimal force applied by the human tongue in the oral cavity [19]. The test vessel was filled with 600 mL of phosphate buffer pH 6.8 at 37 °C (simulated saliva). The samples were cyclically immersed in the buffer, being completely withdrawn from the solution at the highest point and completely submerged at the lowest point of the movement. The endpoint of disintegration was indicated visually when the weights dropped down. Five samples of each batch were measured with results presented as mean values ± SDs.

#### 2.5.5. Mechanical Properties of ODFs

Texture analysis was used to evaluate the mechanical properties of prepared ODFs. A CT3 Texture Analyzer (AMETEK Brookfield, Chandler, AZ, USA) equipped with a 4.5 kg load cell and controlled by TexturePro CT software (AMETEK Brookfield, Chandler, AZ, USA) was used. Films were fixed between two clamps of the TA-DGA probe positioned at an initial distance of 2 cm. The lower clamp was steady while the upper clamp moved at a rate of 0.5 mm/s to pull apart the ODF until breakage occurred. Force and work done during this process, along with the elongation of the film at the point of tearing, were measured. Tensile strength was calculated by dividing the tensile force (TF) at which the film broke by the film’s cross-sectional area (cm^2^). Results of tensile testing (*n* = 5) are presented as mean values ± SDs. The film’s mechanical properties were additionally recalculated to the uniform film thickness of 100 μm to facilitate the comparison [10,23,24].

A texture analyzer with a TA39 cylindrical probe (2 mm diameter, probe motion speed 0.5 mm/s) was used for the puncture test. The force needed to puncture film fixed in the JIG TA-CJ holder, along with the work done during this process and the deformation of the film at the point of penetration, were measured. Results of the puncture test (*n* = 5) are presented as mean values ± SDs. Puncture work (hardness) and tensile work were calculated by TexturePro CT software as an integral of force over distance from test start to the point of puncture or rupture of specimen. Obtained results were also recalculated to a uniform thickness of 100 μm [24].

#### 2.5.6. Statistical Evaluation

Data processing aimed to determine the effect of drying time on selected film characteristics utilizing standard statistical methods. The comparison of all measured values for individual film properties at each thickness level was carried out using a *t*-test. The dependencies in the whole data set for a given film characteristic were investigated via analysis of variance (ANOVA) and paired *t*-test for mean values at each thickness level. Spearman’s correlation coefficient (r_s_) as a robust non-parametric correlation analysis method was used to assess the relationships between RMC and mechanical properties of the films. Statistical significance of the effects is presented as *p*-values (the significance level was set to 0.05). The data analysis was performed using R software, version 4.0.1 (Vienna, Austria) [25].

## 3. Results and Discussion

### 3.1. Physical Properties

#### 3.1.1. Weight and Thickness of Films

Weight uniformity is crucial since the drug content is determined by changes in the shape of the digital model or drug concentration in dispersion. Therefore, weight uniformity is closely related to dose accuracy [12]. Results of ODF weight (Table 3) show only a slight difference between experiments I. and II. (0.25–1.89%). All samples in both experiments exhibited low weight variability (RSD_max I._ = 5.34%, RSD_max II._ = 2.58%) and a strong correlation between the thickness of the digital model (theoretical) and the weight of ODFs (R^2^ = 0.9999 and 0.9997, respectively). The statistical evaluation using paired *t*-test for mean values and comparison individually for each thickness level through *t*-test did not show a significant effect of drying time on ODFs weight (*p* > 0.05 for all cases). Based on these results, it could be concluded that the shorter drying time used in experiment I. is sufficient to achieve an acceptable weight uniformity.

In general, it is recommended that the thickness of orodispersible films should be in the range of 10–100 µm [4]. However, some studies show examples in which the recommended range was exceeded when aiming to improve ODFs’ mechanical properties while maintaining short disintegration times [20,26]. The statistical evaluation using ANOVA and individual comparison of each thickness level through *t*-test did show a significant effect of drying time on the final thickness of the films (*p* < 0.05 for all cases). The thicknesses found ranged from 44.39 ± 5.26 to 221.22 ± 23.98 µm (experiment I.) and from 46.83 ± 5.82 to 211.95 ± 14.09 µm (experiment II.) (Table 3). The film thickness differences between batches ranged from 6.66 to 7.08 µm (3.34–7.78%).

#### 3.1.2. Residual Moisture Content

Residual moisture content in the ODFs is one of the most important parameters. It strongly influences their stability (degradation and undesirable reactions of components and APIs; microbial contamination, etc.) and stickiness (application comfort, packaging). Suitable water content acts as a strong plasticizer and reduces a film’s fragility [21,27]. According to Nair et al. the moisture content in oral film should be lower than 5% [28]. In other studies, the limit was experimentally determined in the range between 3 and 6% [21], respectively, 3.4–6.2% [22]. The moisture content values in experiment I. ranged from 1.05% to 3.47% and from 0.84% to 3.24% in experiment II., respectively (Figure 1, Table 3). This equals the relative net loss of water content between 92.79% and 99.02% when related to the initial water content in printing formulation (86 wt%). In general, the statistically significant effect of drying time on the moisture content in ODFs was proved by ANOVA (*p* = 0.029). The greatest difference could be observed in samples with the middle thickness (percentage difference: 85 µm—40.93%; 125 µm—50.93%; 165 µm—20.23%). Additionally, the *t*-test confirmed a statistically significant influence of drying time on the RMC for thickness levels of 85 µm (*p* = 0.049) and 125 µm (*p* = 0.020). On the other hand, the difference between the RMC values of 45 µm batches was only 6.64% (3.24–3.47% RMC, respectively) while absolute values were considerably higher than in thicker batches. Due to the hygroscopic character of formulation, we hypothesize that outer parts of ODFs may absorb humidity even in the short time between printing and RMC evaluation. All ODFs had a comparable surface area (top and bottom faces account for 2 × 600 mm^2^ regardless of thickness, while side faces add only 4.5–20.5 mm^2^ of the surface area following thickness). Therefore, the amount of moisture absorbed, while the same in absolute values, would affect the relative content of moisture mainly in the thinnest ODFs.

#### 3.1.3. Surface pH

There is a protective buffering mechanism in the oral cavity against irritation caused by changes in the pH. This buffering system is an important property of saliva that also protects against dental caries. Saliva consists mainly of bicarbonate buffer and other buffer systems (phosphate, protein, and mucin), keeping the natural pH of the oral cavity in the range of 5.8–7.4 [29,30]. Nevertheless, it is recommended that the ODFs’ pH values are in this range to avoid irritation of oral mucosa.

The surface pH values of films from experiment I. were 6.9–7.3 (Table 3). In general, these values were slightly lower than in experiment II. (7.2–7.6), while this difference was identified as statistically significant by all statistical approaches used (*p* < 0.05 in all cases). Seemingly, the increase in pH may be primarily driven by a lower RMC. The pH of only one batch of the second experiment (45 µm) was out of the recommended range (Table 3). Irritation of the oral mucosae by these ODFs is not expected due to their low weight [31].

#### 3.1.4. Disintegration

The European Pharmacopeia does not specify the limits or methods for evaluating the disintegration time of ODFs. In practice, limits for orodispersible tablets (ODTs) are used to assess the disintegration time (180 s according to Ph.Eur., 30 s according to FDA, respectively) [19,32]. In general, films should disintegrate or dissolve faster than tablets due to their large surface size (2–8 cm^2^) available for wetting by saliva [33,34].

The films from experiment I. showed slightly faster disintegration times (Table 4). The values of disintegration time ranged from 2.6 ± 0,32 to 40.6 ± 1.37 s. In the second experiment, the values ranged from 2.7 ± 0.17 to 41.3 ± 1.66 s. This is comparable to a study employing a similar printing technique, only with a 6 h long post-print drying time, where the disintegration time found was between 2.02 and 49.85 s [35]. In both experiments, the thickest batches do not meet the specific limit for orodispersible tablets. The differences in the disintegration time between experiments can be explained by lower moisture content in films found in the second experiment. However, according to ANOVA and paired *t*-test, the difference between these experiments is not statistically significant even after recalculating the disintegration time to the uniform thickness of 100 µm (*p* > 0.05 for all cases). RMC influences the structure of the ODFs. Especially, water molecules bound by hydrogen bonds can result in larger pore size formation. When the moisture content is higher, the solvent molecules can easily penetrate the film structure through pores, accelerating the overall disintegration [14,36]. Statistically insignificant differences between the same batches from experiments I. and II. indicate that moisture content has only a partial impact on the disintegration time.

#### 3.1.5. Mechanical Properties

Mechanical properties of orodispersible films are the crucial parameters impacting handling, application comfort, and packaging [37]. Mechanical properties were characterized by puncture testing (puncture force, puncture work, and deformation) and tensile testing (peak load, tensile strength, tensile work, and elongation) [24,38,39].

##### Puncture Testing

The results of puncture testing are summarized in Table 5. The ODFs from the first experiment exhibited lower puncture force and puncture work values but higher puncture deformation values. These films are more plastic and yet more prone to puncture. The films from the second experiment have higher puncture force and puncture work values but lower puncture deformation. Hence the films are harder and less flexible. The above dependencies were confirmed for mean values comparison by ANOVA and paired *t*-test and by comparing the puncture testing parameters at individual thickness levels by a simple *t*-test (*p* < 0.05 for almost all cases). The RMC influences the mechanical properties of films due to its plasticizing properties, following the results obtained (Table 3) [40,41]. A statistically significant negative correlation between RMC and puncture force/work was revealed (Table 5).

Orodispersible films do not have defined mechanical parameter limits. Yet, there are several experiments available for comparison. Preiss et al. inspected marketed orodispersible films, finding puncture strengths between 0.08 and 0.36 N/mm^2^. The conclusion of this study recommended that the films reach at least 0.06 N/mm^2^ [20]. ODFs prepared in the presented experiments meet the recommended limit (Table 5).

Moreover, Brniak et al. accepted orodispersible film puncture strength values ranging from 1.30 to 43.57 N, with the highest tensile strength found in placebo films without active substances [42]. The incorporated materials influence the properties of ODFs more. However, based on the presented data, the properties could be tuned by the drying process as it has a statistically significant effect on the results of puncture testing. the results of both experiments proved that all of the prepared ODFs are sufficiently resistant to be handled safely, and the in-process drying setup is adequate for preparation of ODFs.

##### Tensile Testing

According to relevant literature, orodispersible films reach their peak load at 0.32–0.34 N with an average elongation increase of 14.24% [43]. For sufficient handling properties, the recommended percentual elongation of ODFs at break is >10% [44,45]. However, the elongations vary between experimental studies due to differences in the materials used. For example, only 2.10–3.76% of elongation was found in one of the studies [35].

Further analysis of commercial products shows that the tensile strength should be at least 30 N/cm^2^ [44,46]. The ODFs loaded with quetiapine in a different study show much smaller tensile strength values (2.12–3.65 N/cm^2^) and percentual elongation of 5.01–6.33% [47].

The results of tensile testing are summarized in Table 6. All obtained values in both experiments are above the recommended limits for tensile strength. Experiment I. shows that the tensile strength of films has a decreasing tendency, from the smallest to the largest thickness. In the second experiment, the tensile strength of the ODFs has an increasing but not linear trend. The results from the second experiment show that the ODFs are similarly flexible, but more force is required to tear them compared to experiment I.; they also offer higher tensile strength values. The statistical insignificance of the drying time effect on the deformation at peak load (*p* > 0.05) and, on the other hand, the confirmed significant differences in peak load (*p* < 0.05) and tensile strength (*p* < 0.05) between experiments I. and II. identified by various statistical methods also correspond to this finding. Tensile work also has higher values in the second experiment.

A comparison of the results shows that films from the second experiment are stronger but less flexible, meaning that they can experience higher loads and higher forces until a rupture occurs. As is the case in puncture testing, the residual water content strongly influences the tensile mechanical properties of the films, especially peak load, and tensile work, where the significant negative correlation with RMC was confirmed (Table 6) [3].

## 4. Conclusions

SSE 3D printed ODFs with different drying processes were compared. Statistically significant impacts of drying time on thickness, moisture content, hardness, deformation at hardness, work at hardness, peak load, tensile work, and tensile strength were found. These impacts are caused by the change in moisture content, which affects the physico-chemical properties of the films. Statistical evaluation of ODFs properties showed that weight, disintegration, and elongation differences are not statistically dependent on the drying setup.

The films prepared using the in-process drying met the recommended limits for moisture content. The plasticizing properties of water were preserved, contributing to good mechanical properties. The sole in-process drying is an acceptable drying method.

In conclusion, using different drying setups is a suitable way to fine-tune the properties of SSE printed ODFs without compromising the weight uniformity or disintegration time. However, such tuning comes at the cost of longer preparation times, reducing the effectiveness of this manufacturing method as a tool for the preparation of ODFs.

## Figures and Tables

**Figure 1 pharmaceutics-14-00250-f001:**
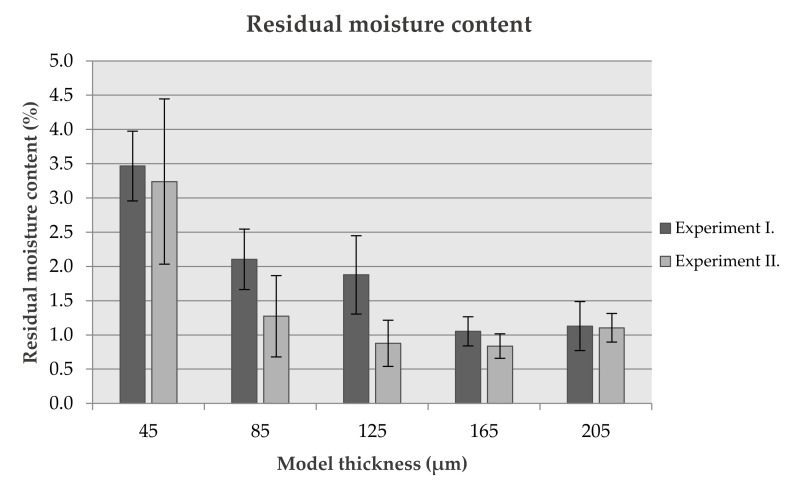
Graph of residual moisture content (%).

**Table 1 pharmaceutics-14-00250-t001:** Printing parameters.

Parameter	First Layer	Other Layers
Height of the layer (mm)	0.005	0.02
The rectilinear pattern infill density (%)	100	100
Perimeter	1	1
Extrusion width (mm)	0.7	0.84
Print speed (mm/s)	7.5	20
Bed temperature (°C)	75	70

**Table 2 pharmaceutics-14-00250-t002:** Drying times used in the experiments.

Experiment I.	45 µm	85 µm	125 µm	165 µm	205 µm
Drying during printing (min)	31	50	68	86	104
Additional drying (min)	10	10	10	10	10
Total drying time (min)	41	60	78	96	114
Experiment II.					
Drying during printing (min)	31	50	68	86	104
Additional drying (min)	83	64	46	28	10
Total drying time (min)	114	114	114	114	114

**Table 3 pharmaceutics-14-00250-t003:** Properties of ODFs from experiments I. and II.

	Model Thickness (µm)	Weight (mg)	Thickness (µm)	Moisture Content (%)	pH
Experiment I.	45	41.18 ± 2.13	44.39 ± 5.26	3.47 ± 0.51	7.08 ± 0.75
	85	74.87 ± 2.11	82.05 ± 5.19	2.10 ± 0.44	6.96 ± 0.08
	125	111.90 ± 5.98	112.19 ± 10.60	1.88 ± 0.57	6.91 ± 0.05
	165	141.47 ± 3.71	158.43 ± 10.43	1.05 ± 0.21	7.33 ± 0.07
	205	183.89 ± 1.99	221.22 ± 23.98	1.13 ± 0.36	6.98 ± 0.02
Experiment II.	45	40.50 ± 0.52	46.83 ± 5.82	3.24 ± 1.21	7.55 ± 0.02
	85	74.88 ± 1.94	86.70 ± 6.64	1.27 ± 0.59	7.34 ± 0.03
	125	111.77 ± 1.81	119.98 ± 14.56	0.88 ± 0.34	7.32 ± 0.03
	165	144.85 ± 1.86	163.07 ± 17.47	0.84 ± 0.18	7.28 ± 0.02
	205	182.38 ± 1.75	211.95 ± 14.09	1.10 ± 0.21	7.23 ± 0.01

**Table 4 pharmaceutics-14-00250-t004:** Disintegration time of ODFs and disintegration time recalculated to a thickness of 100 µm.

	Model Thickness (µm)	Disintegration Time (s)	Disintegration Time Recalculated to a Thickness of 100 µm (s)
**Experiment I.**	45	2.60 ± 0.33	5.80 ± 0.44
	85	7.20 ± 0.36	8.80 ± 0.50
	125	12.30 ± 1.84	10.99 ± 1.41
	165	24.40 ± 0.35	15.38 ± 0.41
	205	40.60 ± 1.37	18.36 ± 0.61
**Experiment II.**	45	2.70 ± 0.17	5.76 ± 0.37
	85	8.10 ± 0.23	9.37 ± 0.26
	125	14.50 ± 0.50	12.06 ± 0.42
	165	24.20 ± 0.70	14.86 ± 0.43
	205	41.30 ± 1.67	19.47 ± 0.78

**Table 5 pharmaceutics-14-00250-t005:** Summary of results from puncture testing.

	Model Thickness (µm)	Hardness (g)	Hardness (N)	Deformation at Hardness/Puncture Deformation (mm)	Puncture Work (mJ)
Experiment I.	45	143.85 ± 12.73	1.41 ± 0.12	4.26 ± 0.57	3.85 ± 0.69
	85	227.11 ± 12.51	2.23 ± 0.12	4.58 ± 0.08	5.46 ± 0.22
	125	228.16 ± 23.79	2.24 ± 0.23	5.18 ± 0.24	6.11 ± 0.56
	165	333.59 ± 9.58	3.27 ± 0.09	5.20 ± 0.11	8.50 ± 0.35
	205	429.63 ± 72.08	4.22 ± 0.71	5.05 ± 0.04	11.69 ± 1.42
Experiment II.	45	157.70 ± 23.58	1.55 ± 0.23	3.24 ± 0.35	4.00 ± 0.50
	85	333.60 ± 21.56	3.27 ± 0.21	3.54 ± 0.18	6.57 ± 0.52
	125	531.20 ± 39.53	5.21 ± 0.39	3.68 ± 0.16	10.27 ± 0.63
	165	708.70 ± 32.50	6.95 ± 0.32	3.68 ± 0.21	13.85 ± 1.00
	205	918.10 ± 38.36	9.01 ± 0.38	3.69 ± 0.18	17.43 ± 1.56
r_s_ (*p*-value) *	-	−0.867 (0.003)	−0.888 (0.001)	−0.024 (0.947)	−0.867 (0.003)

* r_s_—Spearman correlation coefficient between the RMC and the relevant quantity.

**Table 6 pharmaceutics-14-00250-t006:** Summary of the results from tensile testing.

	Model Thickness (µm)	Peak Load (g)	Peak Load (N)	Deformation at Peak Load/Tensile Deformation (mm)	Tensile Work (mJ)	Tensile Strength (N/cm^2^)
Experiment I.	45	229.37 ± 25.78	2.25 ± 0.25	4.89 ± 1.07	14.69 ± 2.22	250.60 ± 30.47
	85	440.56 ± 24.36	4.32 ± 0.24	7.69 ± 1.04	35.88 ± 5.56	258.62 ± 11.19
	125	482.05 ± 46.42	4.73 ± 0.46	8.24 ± 0.87	48.37 ± 6.78	211.75 ± 24.64
	165	738.64 ± 21.43	7.25 ± 0.21	9.63 ± 0.39	82.02 ± 3.73	225.82 ± 7.55
	205	852.01 ± 80.23	8.36 ± 0.79	7.99 ± 1.70	128.98 ± 14.76	195.22 ± 15.31
Experiment II.	45	352.60 ± 19.09	3.46 ± 0.19	3.70 ± 0.55	21.14 ± 3.36	366.62 ± 28.27
	85	749.90 ±10.68	7.36 ± 0.10	8.74 ± 0.58	65.94 ± 4.67	418.79 ± 17.13
	125	1075.30 ± 54.93	10.55 ± 0.54	6.55 ± 0.70	83.53 ± 7.13	440.67 ± 18.32
	165	1421.40 ± 85.61	13.95 ± 0.84	8.33 ± 1.60	133.39 ± 17.87	421.83 ± 32.45
	205	1846.30 ± 72.91	18.12 ± 0.72	9.97 ± 1.16	217.54 ± 15.62	430.75 ± 24.65
r_s_ (*p*-value) *	-	−0.867 (0.003)	−0.867 (0.003)	−0.564 (0.096)	−0.867 (0.003)	−0.418 (0.232)

* r_s_—Spearman correlation coefficient between the RMC and the relevant quantity.

## Data Availability

Data are contained within the article.

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
