# Peer review of "Effects of Various Drying Times on the Properties of 3D Printed Orodispersible Films"

_pharmaceutics, 2022, doi:10.3390/pharmaceutics14020250_

Round 1

Reviewer 1 Report

The paper titled “Effects of various drying times on the properties of 3D printed 2 orodispersible films” aimed at comparing two setups of drying used to prepare ODF by semisolid extrusion. The study evaluated the prepared ODF for their surface pH, disintegration time residual moisture content and mechanical properties. Although the paper is interesting but there are a number of gaps in the experimental design making it difficult to justify some of the results. The paper will benefit from studying the internal structure of the prepared films to help in explaining some of the trends.

The introduction is well written and cover the relevant literature in the area.

The methods are well reported, however it does not provide details of how the puncture work, tensile strength and tensile work was calculated

Results and discussion

Table 3, report the p values rather than reporting 0 and ****

It is not clear why the residual moisture content is low in thick films compared to thin filmed, one assumes that evaporation will be more efficient from thin surfaces. Possibly a study of the internal structure of the films can help explain this?

The authors claim that the surface area for all ODF is the same despite formulations have different thickness.

For the mechanical properties, it might be good if the author evaluate any correlation between the RMC and the tensile strength or the puncture strength.

Reviewer 2 Report

In this manuscript, the authors are trying to investigate the influence of drying times on the properties of 3D printed films using semi-solid extrusions. This is an interesting manuscript that addresses an important objective as the drying time will have a direct impact on the residual moisture content and the quality of the films. The authors have investigated the influence of a fixed drying time to a variable drying time depending on the processing times for the films and their effect on the weight, thickness, residual moisture content, pH, texture, and disintegration time of the manufactured films. Even though the objective and rationale of this study are important, there are some flaws in the study design that should be revised before publication.

  • The authors in this manuscript are preparing placebo films without any drug component. Why did the authors select a placebo formulation?
  • As the authors correctly mentioned in the introduction, the drying times can influence the solid-state of the drug and can induce recrystallization of the drug as undesired polymorphs making the performance of the dosage form unpredictable as they have different physical and chemical properties, why are the authors not investigating the effect of drying time on the solid-state of the drug i.e., melting point, crystalline form, amorphous conversion, degradation, etc.?
  • Why did the authors choose thermogravimetric analysis for moisture content instead of Karl-Fischer which is more sensitive for the measurement of moisture content? With thermogravimetry there are chances that the weight loss is due to impurities or degradation of components in the formulation, to eliminate any sources of error the authors will have to analyze each component separately to ascertain that there is no degradation at the analysis temperature and the weight loss is solely due to the loss of water.
  • What is the total moisture/water content in the ink pre-processing? (Even though this is something that can be calculated from the formulation composition, assessing this will provide the net loss of water post-processing which can be related to the total drying time.)
  • Did the authors consider determining moisture content over different drying times to calculate the drying rate during the process? This way the drying rate can be used to achieve a desired residual moisture content at the end of the process, this will make the present study more impactful.

Minor comments

  • The introduction is well written.
  • Line 43: Please elaborate and support this statement with proper citations “ODFs can impact drugs' therapeutic effects”.
  • Line 90-95: Requires citations.
  • In separate instances, the authors have mentioned that the thickness correlates with the weight of the film, and the drying time had a significant effect on the film thickness, so why is the drying time not influencing the weight of the films? Please elaborate.
  • For the 85 and 125-micrometer thickness films, the moisture content is significantly different between the two drying times as the drying time in method two is longer as compared to method 1, why is this trend not observed in 45 micrometer thick films as the difference between the two drying times were maximum amongst all the batches?
  • Why would the drying process have an influence on the pH of the films? are there any proton donors or acceptors present in the formulation which would change the pH of the microenvironment on dissolution?
  • In line 306 the authors have mentioned “Especially, water molecules bound by hydrogen bonds can result in larger pore size formation.” Have the authors conducted SEM or other morphology studies to support this? Showing the difference in the macroporosity of the films between two different drying processes with variable moisture content would increase the impact of the presented research.

Round 2

Reviewer 1 Report

The authors addressed most of my comments, although I feel more work should be done to investigate the internal structure of the prepared films but this could be something that the authors could consider in the future work. I am happy with the manuscript in its current form.

Author Response

Dear Reviewer, 

Thank you for the suggestions and the opportunity to improve the quality of our work. We are glad that you are satisfied with the manuscript. We will integrate some structural studies in the next experiments, as mentioned in previous comments.

Reviewer 2 Report

The authors have stated in their response that this experiment aimed to relate the drying setup to the properties of the general ODF matrix and that the incorporation of the drug has a tremendous effect on the properties of ODFs, which are strongly related to the type and amount of the selected drug. The speculation that the properties, results, and conclusions would differ significantly from the general ODF matrix on incorporating the drug significantly reduces the importance of this present work as it would not have a correlation with drug-loaded ODFs according to the authors. In my opinion, the placebo batch is critical for this study, however, the study is incomplete without a drug-loaded sample set. The authors are correct in stating that there is a wide range of model drugs that can be used, however, this can be resolved by using a model drug commonly delivered as an ODF. I request the authors to reconsider their responses to my previous comments as their current solution to conduct the studies in their future papers is not a satisfactory response.

Author Response

Dear Reviewer, 

Thank you for the comment. We try to explain it as best as we can. This study was aimed to evaluate the influence of one of the most important process parameters of SSE 3D printed ODFs - the drying process, on the films' properties. Incorporating a model drug could conceal the general findings resulting from this study. That is also why we did not try to speculate the possible influence of a model drug in the manuscript and focused only on the film matrix properties related to the drying setup. The influence of the drug on the properties of the ODFs is based on long-term experience with oral films' research (e.g., https://www.hindawi.com/journals/bmri/2015/580146/ or https://link.springer.com/article/10.1208/s12249-018-1088-y). Based on the obtained results (presented in the manuscript), we will be able to tune the process of drug-loaded ODFs drying in the next study from the point of view of drug influence on the films properties as well as the modification of drug state (crystallinity, etc.) based on drying setup. This is the aim of the follow-up study. We want to examine the incorporation of different drugs in different concentrations from the point of view of the dosage form and the physico-chemical state of drugs. This is a relatively extensive and complex study that cannot be added to this manuscript due to limited time for the review process and the fact that this would substantially change the concept of the whole manuscript.